# Green Methods for the Fabrication of Graphene Oxide Membranes: From Graphite to Membranes

**DOI:** 10.3390/membranes13040429

**Published:** 2023-04-13

**Authors:** Alessandro Pedico, Luisa Baudino, Anna Aixalà-Perelló, Andrea Lamberti

**Affiliations:** 1Politecnico di Torino, Dipartimento di Scienza Applicata e Tecnologia (DISAT), Corso Duca degli Abruzzi, 24, 10129 Torino, Italy; 2Istituto Italiano di Tecnologia, Center for Sustainable Future Technologies, Via Livorno, 60, 10144 Torino, Italy

**Keywords:** graphene oxide, membranes, green fabrication, sustainable processes

## Abstract

Graphene oxide (GO) has shown great potential as a membrane material due to its unique properties, including high mechanical strength, excellent thermal stability, versatility, tunability, and outperforming molecular sieving capabilities. GO membranes can be used in a wide range of applications, such as water treatment, gas separation, and biological applications. However, the large-scale production of GO membranes currently relies on energy-intensive chemical methods that use hazardous chemicals, leading to safety and environmental concerns. Therefore, more sustainable and greener approaches to GO membrane production are needed. In this review, several strategies proposed so far are analyzed, including a discussion on the use of eco-friendly solvents, green reducing agents, and alternative fabrication techniques, both for the preparation of the GO powders and their assembly in membrane form. The characteristics of these approaches aiming to reduce the environmental impact of GO membrane production while maintaining the performance, functionality, and scalability of the membrane are evaluated. In this context, the purpose of this work is to shed light on green and sustainable routes for GO membranes’ production. Indeed, the development of green approaches for GO membrane production is crucial to ensure its sustainability and promote its widespread use in various industrial application fields.

## 1. Introduction

Membranes are key components in many industrial processes, such as water treatment, gas separation, and food processing [1,2,3,4,5,6]. They play a crucial role in separating different components in a mixture by selectively allowing certain molecules to pass through while rejecting others based on their size, shape, charge, or other physical properties. Membranes offer several advantages over traditional separation methods such as distillation and filtration, including high selectivity, low energy consumption, and compact size. For example, in water treatment, membranes can remove bacteria, viruses, pharmaceuticals, and dissolved salts from the water, producing high-quality drinking water [7,8,9,10,11]. In gas separation, membranes can separate gases such as oxygen, carbon dioxide, and nitrogen, which are important in various industries such as chemical manufacturing and medical gas production [5,6,12,13,14]. In material recovery, membranes find application for their capability of selectively separating molecules or chemical species [15,16,17,18].

However, the production of membranes can have a significant environmental impact, primarily due to the use of energy-intensive processes and high-impact materials. Membranes are often made from synthetic polymers derived from non-renewable resources, such as petrochemicals. The manufacturing process also consumes a lot of energy and generates waste products. To reduce the environmental impact of membranes, there is a need for green approaches to membrane design, production, and reuse [19,20]. This approach involves using sustainable materials as well as reducing the impact of solvents and developing more energy-efficient manufacturing processes. Recycling and reusing membranes can also help reduce waste and conserve resources [21].

Green membranes can also provide new opportunities for sustainable and circular economies. For example, by using biomass-derived materials to produce membranes, waste products from agriculture and forestry can be turned into valuable materials for industrial applications. Membrane-based processes can also be integrated with other sustainable technologies, such as renewable energy sources and carbon capture and storage [22,23].

Graphene oxide (GO) membranes are an innovative type of membrane that can offer multiple advantages over traditional membranes. GO is a two-dimensional material composed of graphene sheets functionalized with oxygen-containing groups such as hydroxyl, epoxy, and carboxyl groups, giving it unique properties such as high selective permeability and chemical stability.

GO membranes can be used in various applications, such as gas separation [24,25,26], water desalination [27,28,29], wastewater purification [30,31], and biofuel production [32,33]. In these applications, GO membranes can offer advantages such as higher efficiency, selectivity, and durability compared to traditional membranes. Moreover, GO is a sustainable material that can be produced from renewable sources such as cellulose [34] and lignin [35], reducing the environmental impact of membrane production.

In general, GO membranes represent an interesting alternative to traditional membranes and could offer significant environmental and economic benefits in the future, but they are still under study and development, and their industrial-scale application requires further research and development. Additionally, GO is still a costly material compared to other membrane options, which limits its commercial application. For this reason, there is a need for new production processes for low-cost GO membranes.

Membranes are crucial in many industrial processes, and their importance will only increase as the demand for sustainable solutions grows. Particular attention should be paid to the greener production of both GO and GO membranes, making their application in large-scale scenarios possible. The aim of this review is therefore to address the sustainability of all the processes involving the production of graphite oxide to the fabrication of GO membranes. Indeed, a green approach to membrane design, production, and use is essential to reducing the environmental impact of membranes and creating a more sustainable future.

## 2. Graphene Oxide Production

In this section, the production methods to oxidize graphite into graphite oxide and then exfoliate it into GO will be analyzed. Furthermore, different modifications of GO will be taken into account, i.e., reduced GO and functional GO, as those are the most commonly used in membrane technology.

### 2.1. Graphite Oxide

The first productions of graphite oxide are reported by Brodie [36], in 1859, and by Staudenmaier, who optimized the process in 1898 [37]. Brodie was able to obtain graphite oxide by adding potassium chlorate (KClO_3_) to a dispersion of graphite in fuming nitric acid (HNO_3_). Staudenmaier, instead, used a mixture of concentrated sulfuric acid (H_2_SO_4_) and fuming HNO_3_, and then slowly added the KClO_3_. Both obtained a final material with a C:O ratio close to 2:1, but the process developed by Staudenmaier proved to be more reliable and led to more reproducible results. A modification of this process was later described by Hofmann and Konig in 1937 [38]. The synthesis of graphite oxide was simplified by using concentrated HNO_3_ instead of the fuming one, thus becoming potentially scalable. However, this process never reached an industrial scale because of the long time required for the oxidation of graphite.

A different and much faster approach for the oxidation of graphite was proposed in 1958 by Hummers and Offeman [39]. Graphite was first added to a solution of H_2_SO_4_ and sodium nitrate (NaNO_3_), to which potassium permanganate (KMnO_4_) was then slowly added. Finally, a solution of hydrogen peroxide (H_2_O_2_) was employed to wash the suspension, resulting in a material with a C:O ratio slightly lower than 3:1. This method proved to be extremely successful and is now known as the “Hummers’ method” (see Figure 1). The “Hummers’ method” is the basis of all the modified versions later proposed. The reason behind its success and large diffusion is related to the possibility of obtaining graphite oxide in a few hours, compared to the few days of the previously proposed methods.

Although very rapid and high-yielding, the Hummers’ method presents several drawbacks in terms of the toxicity and safety of the process. Not only does it involve the formation of toxic gases, as did the previously proposed methods, but it also presents different chemical hazards related to its exothermicity and the formation of explosive compounds [41]. Several “modified Hummers’ methods” have then been proposed over the years to reduce the impact of this process. To avoid the formation of toxic gases such as nitrogen dioxide (NO_2_), several studies have proposed the substitution of NaNO_3_ with other compounds. For example, Marcano et al. used phosphoric acid (H_3_PO_4_) in 2010 [42], while Liu et al. used potassium persulfate (K_2_S_2_O_8_) and phosphorous pentoxide (P_2_O_5_) in 2012 [43]. In 2016, Yu et al. discarded the use of NaNO_3_ and partially substituted KMnO_4_ with K_2_FeO_4_, thus reducing the number and amount of hazardous byproducts [44]. Different methods were also studied to obtain a safer process, such as the electrolysis of graphite rods proposed in 2018 by Aslam et al. [45]. Finally, the role of the temperature was also closely investigated in several studies in light of the exothermicity of the oxidation reaction. In 2019, Al Gaashani et al. tested the influence of temperature in the presence of a mixture of H_2_SO_4_, H_3_PO_4_, and HNO_3_, and then KMnO_4_ and H_2_O_2_ without the use of other oxidizing agents [41]. Finally, in 2022, Mendez-Lozano et al. reported the synthesis of graphite oxide using concentrated H_2_SO_4_ and KMnO_4_ and later H_2_O_2_ [46]. By using a two-step temperature process, they reported a final material similar to the one obtained through the Hummers’ method without the need to use NaNO_3_ or other oxidizing agents.

### 2.2. Graphene Oxide

GO has attracted a lot of attention due to its interesting properties, such as its large surface area, great mechanical strength, electrical conductivity, chemical inertness, and ease of functionalization [47,48]. While the methods previously described start with pristine graphite and oxidize it into graphite oxide, a further exfoliation step is required to obtain GO. The two most common techniques to induce mechanical exfoliation to single- or few-layer GO are sonication [49] and thermal shock by a rapid cooling process [50]. A combination of the two processes is also possible, as suggested by Aksoy and Anakli [51]. In this case, the dispersion of GO was obtained by applying a potential difference to graphite electrodes in a thermally heated sonication bath. Ball milling has also been investigated to exfoliate graphite oxide. Nevertheless, the quality and the degree of oxidation are highly affected by the solvent, with mainly reduced graphene oxide achieved when using water [52].

Given the numerous available options to obtain GO, the sustainability of the process is of fundamental importance. Taking the chemical route into consideration, hazardous chemicals should be avoided for both safety reasons and their environmental impact [41]. Not only the reagents themselves but also the byproducts of the reactions should be carefully monitored and managed. Generally speaking, GO preparation methods involving nitrates should not be considered “green”. As previously highlighted for the graphite oxidation processes, the degradation of HNO_3_ and NaNO_3_ can lead to the formation of toxic and polluting gases such as NO_2_ and dinitrogen tetroxide (N_2_O_4_), the latter of which is also explosive. Another explosive gas that can be created as a byproduct in the degradation of KClO_3_ is chlorine dioxide (ClO_2_), and the highly volatile and explosive manganese heptoxide (Mn_2_O_7_) can also arise when KmnO_4_ is subjected to temperature fluctuations. Moreover, these harsh acidic conditions lead to highly damaged structures and the presence of impurities in the GO sheets [53]. Considering instead a non-chemical route, the electrolysis of graphite rods seems a promising green GO production method [45,53,54,55]. This alternative technique achieves high-quality GO with a controlled oxidation degree, a low density of defects, and no impurities. However, the amount of energy required for mass-producing GO through this process can be extensive; therefore, a careful evaluation of the impact of this process should be carried out to evaluate whether it should be preferred to the chemical route.

### 2.3. Reduced Graphene Oxide

GO has attracted a lot of attention and is employed in a lot of applications. However, its reduced form, called “rGO”, may be more widely used and studied in light of its electrical, chemical, and physical properties. The fact that rGO can maintain its exfoliated structure while reducing the number of functional groups on its surface makes it a very promising and versatile material for several different applications [56]. Different reduction strategies have been proposed in the literature, ranging from thermal reduction to chemical reduction [57]. The reduction treatment can be achieved through thermal annealing up to several hundred degrees to remove the oxygen-containing functional groups. The temperature employed will greatly influence the reduction degree and the chemistry of the removed functional groups [58,59,60,61]. However, particular care must be taken with thermal annealing as the reduction of the functional groups could lead to the formation of carbon monoxide and dioxide gases between the flakes, thus inducing structural damage to the stacked flakes [62]. Alternative reduction mechanisms involve exposure to UV light [63,64] or microwaves [65,66]. rGO can also be obtained in the form of an aerogel through a hydrothermal treatment during which a GO water-based solution is exposed to high pressure and temperature (typically 180 °C in a sealed hydrothermal reactor) [67].

The chemical reduction of GO into rGO was first proposed in 2009 by using hydrazine (N_2_H_4_) [68], which although a strongly pollutant and later discovered to be carcinogenic, is still widely used [69,70,71]. Since then, other approaches have been proposed, but most of those currently used require the use of strong acids such as hydrogen iodide (HI) [72,73,74,75,76] and bases such as sodium borohydride (NaBH_4_) [68,77,78,79,80]. However, greener approaches have also been reported lately using ascorbic acid, i.e., Vitamin C, and other natural reducing agents [57,81,82]. Finally, an electrochemical reduction of GO can also be performed with either a current-controlled or a voltage-controlled method. In this way, an rGO-coated surface can be directly obtained, but the process is limited to conductive materials to be used as electrodes [83].

### 2.4. Functionalized Graphene Oxide

GO has also attracted the interest of the scientific community for the possibility of tuning its functional groups instead of removing them and obtaining rGO. The umbrella term “functionalized GO” (fGO) is then used for all the modifications that can be performed on pristine GO. Covalent and non-covalent strategies can be applied to modulate their structure and intrinsic properties [84]. GO can be functionalized by replacing its oxygen-containing functional groups, doping its structure with different atomic species [85,86,87,88], decorating it with nanoparticles [89,90,91,92,93], grafting polymeric chains [63,94,95,96,97], or any combination of the previous. GO sheets are of particular interest as nanoparticle supports due to their high surface area [84]. Instead, functionalization with organic compounds is being studied to improve its solubility and processability in water or organic solvents to take advantage of both materials [84]. GO-polymer nanocomposites, instead, are of great interest for electrochemical applications and to reinforce the structure of the membrane [84]. It must also be noted that two different kinds of functionalization processes can be performed on GO membranes. While the most intuitive one would be the modification of the GO flakes in solution before the realization of the membrane, it is also possible to directly modify the surface of a GO membrane by grafting a superficial layer or immersing it in an activating solution [98,99]. The choice of GO functionalization eventually depends on both its chemistry and field of application.

## 3. Graphene Oxide Flakes Properties

### 3.1. Stacked Structures

GO membranes consist of a stack of GO sheets in laminated structures that create nanochannel galleries. This structure remains integral thanks to the π–π bonds in the sp^2^ clusters and hydrogen bonds in the sp^3^ regions. The voids between the GO flakes will instead shape the channels in which the different species will diffuse. These channels, of sub-nanometric size, are comparable to the size of ions and molecules and are responsible for interesting properties such as high ionic conductivity, ultrafast mass transport, and an anomalous increase in capacitance [100]. The structure of these water channels, which is influenced by the interlayer distance between the flakes, the dimension of the GO sheets, and the charge of the starting material (see Figure 2), will determine the properties and the best application for each membrane [101,102,103]. In particular, the interlayer distance between the flakes will determine the cut-off molecule dimension, whereas the size of the nanosheets will instead affect the length of the channels and the number of inlets on the surface of the membrane. While larger flakes will enhance separation properties by lengthening the percolating path, smaller ones will promote permeation. Finally, the charge density of the flakes will determine the interaction of the membrane with charged molecules [103,104,105].

### 3.2. GO and rGO Dispersion Behavior

Apart from a highly laminated structure of GO, good dispersibility of GO is required for facile fabrication of the membranes [106]. Flake agglomeration should be avoided to achieve a homogeneous structure. However, GO solubility varies not only depending on its oxidation state but also on the solvents used and can be time-dependent. GO presents good dispersibility in NMP (*N*-methyl-2-pyrrolidone), DMF (*N*,*N*-dimethylformamide), water, and ethylene glycol. Non-polar solvents such as toluene, chlorobenzene, and o-DCB (o-Dichlorobenzene) also present low but stable dispersibility. rGO, instead, presents good dispersibility in NMP, water, and ethylene glycol, thanks to the oxygen functional groups that remain in their structure. Additionally, greater interactions with non-polar solvents (chloroform, toluene, and chlorobenzene) are achieved with rGO [107]. Almost all the above-mentioned solvents present different hazards, such as toxicity (NMP, DMF, ethylene glycol, toluene, chlorobenzene, and chloroform), irritation (NMP, DMF, toluene, and chloroform), flammability (DMF, toluene, and chlorobenzene), long-term aquatic hazards (toluene and chlorobenzene), and carcinogenicity (chloroform). The most promising alternative solvent for green GO preparations is water. The solubility in water of both GO and rGO depends on the electrostatic repulsion due to the negative charge of the material [108]. However, water-based GO solutions are not always possible to prepare, depending on the kind of functionalization performed on GO flakes [63]. Another situation in which the dispersion is not straightforward is when working with insoluble binders in composite membranes [109]. Usually, sonication is used to facilitate the dispersion of GO flakes, which can be achieved even in low boiling point solvents by tuning the sonication time and power (see Figure 3) [110]. However, long sonication times can induce defects and reduce the flake size, thus changing the properties of the final material [111]. Overall, when using water is not possible, solvents such as ethanol and DMSO combined with short sonication times represent the most environmentally friendly alternative.

### 3.3. Mechanical Stability and Swelling

The stacked structure of GO flakes will undergo some changes when immersed in water, as water molecules rapidly permeate due to the low friction path through the non-oxidized nanochannels and tend to slow down in the oxidized ones [112,113,114,115]. In particular, water molecules trapped in the structure will enlarge the membrane channels, causing an overall swelling and weakening of the hydrogen bonding between the flakes, thus reducing the mechanical stability of the membrane [116,117]. Even though GO membranes in a dry state present an interlayer spacing of ~9 Å, when soaked in water, due to their hydrophilicity [118], the interlayer spacing can increase up to ~13 Å [119,120]. This can result in rapid ion transport but also in a reduction of selectivity [121,122]. Furthermore, since negatively charged moieties of GO and water molecules repulse each other, this can ultimately result in the dispersion of GO in the aqueous solution [113,123].

The nature and number of functional groups on the GO flake’s surface will also greatly impact its stability in aqueous solutions. While hydroxyl groups promote the presence of hydrogen bonds between flakes and therefore facilitate their cohesion, carboxylic groups can be easily hydrolyzed and contribute to the hydrogen bonds only when protonated at low pH [124,125].

Several different strategies have been proposed to reduce swelling, both with chemical and physical methods [101]. Among the chemical methods proposed, one must mention the intercalation of cations and of in situ nanoparticles, the reduction of GO flakes, and the use of cross-linkers. The use of cations and nanoparticles to finely tune the interlayer distance has been extensively studied for ion sieving and ion filtration applications (see Figure 4a,b) [99,113,126,127]. Of particular interest is the fact that multivalent cations show stronger interactions with the GO flakes and therefore ensure better stability of the stacked structure in aqueous solutions [25,113,117] up to the creation of hydrogel structures (see Figure 4c,d) [128]. However, the presence of cations that could re-dissolve into the solution could also cause contamination during the final product operations [117,129]. The incorporation of in-situ grown particles via solvothermal reactions or ALD has also been proposed [26,130]. However, these processes often require the use of either toxic precursors or toxic solvents, such as DMF.

As was previously explained, a partial reduction of the GO flakes can also enhance their stability in aqueous solutions by restoring part of the sp^2^ carbon network. For these purposes, mild annealing should be preferred, and their stability was also shown to improve in strong acids and bases [27,58,102,131].

Finally, a wide range of cross-linking agents has been investigated. Several nitrogen-containing compounds have been studied as cross-linkers, ranging from amines [24,98,132,133,134,135] to diisocyanate [136] to urea and derivatives (see Figure 5a,b) [137,138]. However, only urea can be considered a green cross-linker among the nitrogen-containing compounds. Several organic acids have also been used [112,139,140]. Among the many polymeric chains that have been reported in the literature [25,141,142,143,144,145,146,147], only a few can be considered acceptable in a sustainable approach due to their biocompatibility and safety, i.e., polyethylene oxide (PEO) [25], boronic acid polymer (BA) [142], poly(ethylene glycol) (PEG) [147], poly(methyl methacrylate) [144], and poly(*N*-isopropyl acrylamide) [145].

Physical intercalation and physical confinement have been proposed as physical methods for reducing the swelling phenomenon. Physical confinement has been attempted by using epoxy resins to mechanically restrict the swelling [148]. However, given the impact of epoxy resins, this method would not be suitable for green applications. Physical intercalation instead takes advantage of non-covalent π–π and hydrogen bonding interactions between different nanomaterials [104,149,150,151] or polymers [144,152,153,154,155] and the graphene oxide flakes (see Figure 5c). Production of nanomaterials is moving toward more sustainable and environmentally friendly methods. Nevertheless, the potential environmental impact is still being studied. Among all, PVA, PVP, PEG, and chitosan can be considered safe and biocompatible polymers, even if the production processes and environmental impacts of their disposal are still under investigation.

## 4. Graphene Oxide Membranes Fabrication

Graphene oxide membranes can be separated into two categories, i.e., supported membranes and self-standing membranes [156]. Self-standing membranes only comprise active materials and, depending on the design and structure, can show an increase in flow rate because of the absence of one or more unnecessary layers. They should also be cheaper, as no additional support is needed. However, supported membranes can be easier to fabricate and by wisely choosing both the support and the fabrication method, one can obtain interesting combinations of properties.

The choice of a suitable solvent is one of the main issues in preparing GO membranes in a sustainable way. As previously stated, the dispersibility of GO in aqueous and organic solvents is influenced by many factors, among which its oxidation state and the nature of its possible functionalization are the most important. However, the fabrication method can also influence the choice of solvent for the preparation of GO membranes. The most commonly used solvents for membrane preparation are water, DMF, DMSO, *N*-Methyl-2-pyrrolidone (NMP), ethanol, and isopropyl alcohol (IPA). Generally speaking, when working with GO dispersions, water-based solutions should be preferred. Otherwise, DMSO, ethanol, and IPA solutions can also be acceptable in environmentally friendly fabrications.

There are several different techniques to produce GO membranes, each with its advantages and drawbacks [106,157]. In the following, the most common ones will be analyzed and compared in terms of impact, properties achieved, and scalability.

### 4.1. Layer-by-Layer Assembly

Layer-by-layer assembly is a thin-film technology in which a continuous film of material is obtained by stacking on each other single layers of two different materials. The stacking is controlled by the chemistry of the two materials. Each material spontaneously reacts with the surface of the other material, forming a covalently bonded monolayer. For what concerns the production of GO membranes, this technique has been proposed to improve the stability in water through covalent bonding of the GO sheets [102,123]. Being that the GO sheets are rich in oxygen-containing groups, this is easily achieved using chemical species that spontaneously react with those groups. By repeating the layer-by-layer assembly process, a thick GO membrane can be formed. The main drawback of this technique is the time required to obtain a thick film. For every layer, curing and washing steps are usually required, making the process sub-optimal for more than a few layers of material.

### 4.2. Drop Casting

The drop casting method is the simplest technique to produce GO membranes and allows for the preparation of membranes with various shapes, sizes, and thicknesses [158,159,160]. This technique involves depositing a droplet of a GO solution onto a substrate (see Figure 6a,b). The substrate can be made of different materials, depending on the kind of membrane one wants to obtain. When preparing a self-standing membrane, one can use any flat surface from which the membrane can be easily peeled, such as, for example, glass. Otherwise, in the case of supported membranes, the solution will be cast directly on the polymeric layer that will act as the final support. The solution will then be dried, either at room temperature or under a heating source, thus creating the stacked morphology of GO flakes typical of these membranes. This method can be coupled to a roll-to-roll apparatus for continuous production, achieving the scalability required for industrial applications [161].

### 4.3. Spray Coating

Spray coating is a process used to create thin films with uniform thickness and homogenous properties. This method allows the coating of large-area GO membranes directly onto a substrate surface in a short amount of time. It involves spraying a diluted GO solution onto the surface in a thin, even layer (see Figure 6c) [164]. A nozzle and a pressurized system are required to obtain the spray, while the substrate must be heated and the temperature must be kept constant to allow a uniform and fast evaporation of the solvent. The spray coating method can be used to produce GO membranes with a wide range of thicknesses and properties that can be tailored to specific applications [162,165]. Similar to drop casting and doctor blades, spray coating is also an easily scalable method suited for industrial applications.

### 4.4. Spin Coating

Spin coating is a very fast process that is usually coupled with heating to obtain the desired thin film in a short time [163]. It involves depositing a thin layer of GO solution onto a substrate by spinning it at a high speed (see Figure 6d). This process can be used to produce uniform, thin films of GO with controlled thickness and morphology by tuning the rotation speed, the overall time of the process, and the density and amount of GO solution employed.

### 4.5. Vacuum Filtration

Vacuum filtration is a technique widely used to fabricate GO membranes [28,120,166,167,168,169,170]. This method uses a vacuum pump to create a pressure differential between the two sides of a membrane, which allows the liquid to pass through while retaining the solid particles. To perform the vacuum filtration, a filter funnel is fitted with a membrane and placed on top of a receiving flask. The GO dispersion to be filtered is poured into the funnel, and a vacuum pump is connected to the receiving flask. The vacuum generated by the pump draws the liquid through the membrane and into the receiving flask while the GO flakes stack on each other on the supporting membrane (see Figure 7). At the end of the process, a GO membrane is formed on top of the supporting membrane. Depending on the final thickness of the GO layer and the nature of the supporting membrane, this process can require a higher vacuum level and a longer time. For this reason, hydrophilic membranes with large pores are preferred.

### 4.6. Pressure-Assisted Filtration

Pressure-assisted filtration is a technique involving the use of a pressure source to force a liquid to pass through a membrane while retaining the solid particles [95,103,134,171,172,173]. Similarly to what happens with vacuum filtration, the apparatus commonly consists of a tank connected to a pressure source and filled with the solution. At the bottom of the tank, a membrane is placed. The pressurized gas forces the water to pass through the membrane, while the liquid exits from an outlet. As for vacuum filtration, the driving force is the pressure difference between the two sides of the membrane. If a GO solution is poured inside the tank, an analogous result is obtained, with the GO flakes stacking on the surface of the support membrane, forming the GO membrane. The pressure-assisted filtration is reported to be a valid alternative to vacuum filtration and to produce membranes even more ordered than those prepared by vacuum filtration (see Figure 7).

### 4.7. Doctor Blade

The doctor blade method is one of the methods most commonly used to produce GO membranes and is more promising for scaling up applications. This method consists of coating a substrate with a thin film of GO, which is then dried at ambient temperature or by heating. The GO membranes produced by this method range from ultra-thin to a few microns in thickness. The doctor blade method involves the use of a GO gel that is cast on a flat surface by a rod, or a blade, moved at a constant speed and constant height to uniformly spread it (see Figure 8). The membrane obtained in this way can be peeled off and used either self-standing or supported. The doctor blade method is quick, relatively inexpensive, and simple to use; it is thus a popular method to produce GO membranes [113,174]. For these reasons, the doctor blade method has been proposed as a method to be used on an industrial scale [25].

Different factors affecting both the economic and environmental impact of the membrane should be considered when selecting its fabrication method. The energy and time consumed, for example, will affect the cost of the membrane fabrication. Spray coating followed by drop casting, doctor blades, and spin coating are the most convenient and also scalable techniques. However, these techniques offer less control over the order of the stacked flakes structure of the membranes. Pressure-assisted filtration, followed by vacuum filtration, and layer-by-layer assembly are the most recommended techniques if one needs to prioritize the order of the structure. On the other hand, thickness control and availability are of great interest, as they play an important role in the membrane’s performance (i.e., selectivity or resistance). In fact, if thicker membranes are needed, only drop casting, doctor blades, and pressure-assisted filtration are of interest. Overall, the choice of the fabrication method will also be influenced by the final application of the membrane. A comparison of these characteristics for each method considered is reported in Table 1.

## 5. Conclusions and Perspectives

In recent years, the research on GO membranes has focused on improving their mechanical and selectivity properties. While this is necessary for their large-scale development, the environmental impact of their production cannot be neglected either. Low-energy processes should be preferred to high-temperature or high-energy processes to reduce their environmental impact, and the use of green solvents and reagents also should be implemented. However, the application can limit the amount of sustainability of the process. Whereas in the case of small-scale and precision applications such as sensors, a material that is not entirely “green” can be considered acceptable, this cannot be accepted in large-scale applications. In the case of membranes, the use of materials hazardous to the environment cannot be tolerated anymore, whether they be employed in their synthesis, employment, or disposal. In this context, GO membranes cannot be exempted from undergoing the same considerations. GO membranes are a promising solution for separation technologies, provided that the GO itself and its membrane form can be produced in a sustainable way. For these reasons, we hope that this work will help shed light on green and sustainable routes for GO membranes’ production. In a world in flux, where sustainability has become a dogma, there will be no room for non-green solutions with high environmental impact.

## Figures and Tables

**Figure 1 membranes-13-00429-f001:**
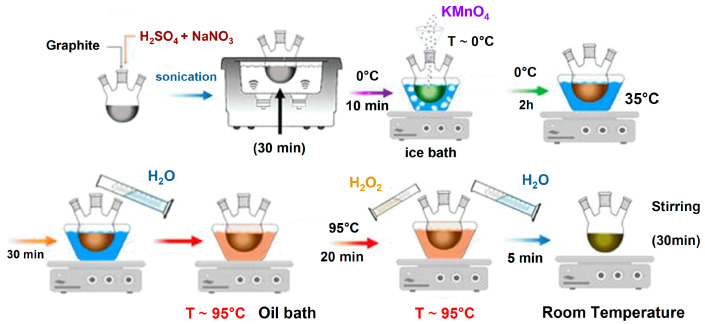
Graphical representation of the steps for obtaining graphite oxide from graphite through the Hummers’ method. Adapted under the terms of the CC-BY license from [40]. Copyright 2020, the authors, published by IntechOpen.

**Figure 2 membranes-13-00429-f002:**
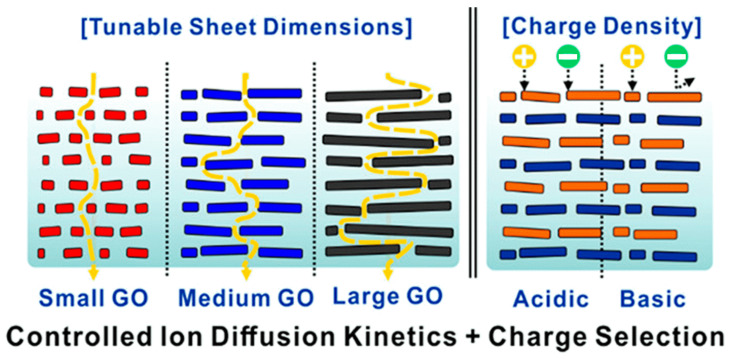
Schematic representation of the ion transport through multilayered GO membranes with differently sized GO building blocks. Reprinted with permission from [102]. Copyright 2020 American Chemical Society.

**Figure 3 membranes-13-00429-f003:**
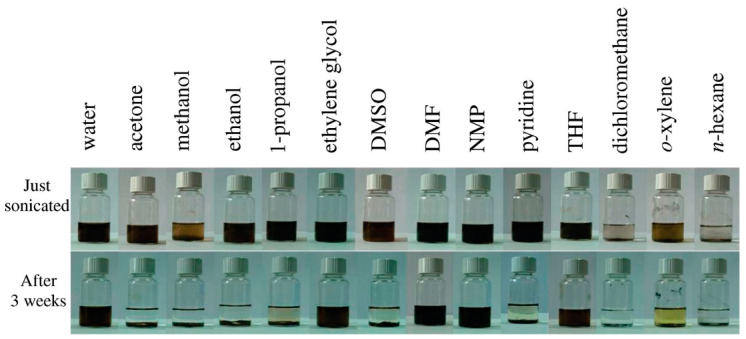
GO dispersion in a range of solvents immediately after sonication (**top**) and after settling for 3 weeks (**bottom**). Reprinted with permission from [110]. Copyright 2020, American Chemical Society.

**Figure 4 membranes-13-00429-f004:**
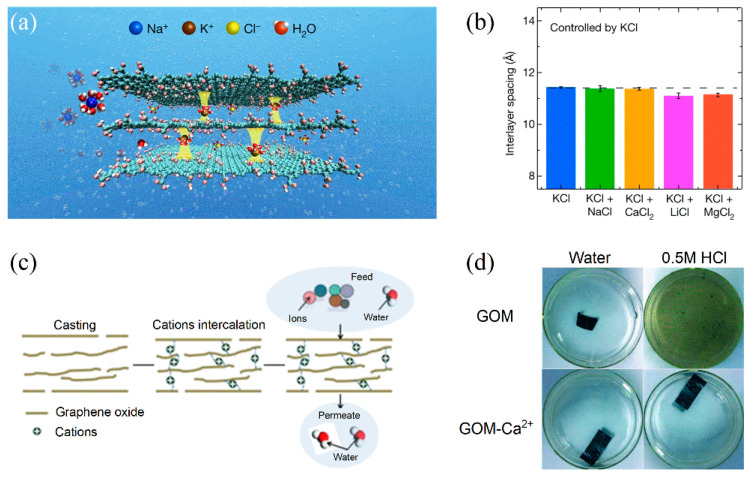
Cation-controlled GO membranes. (**a**) Interaction between K^+^ ions and GO flakes; (**b**) interlayer spaces of wet membranes when controlled by K^+^ ions. Adapted with permission from [127]. Copyright 2017, Springer Nature; (**c**) scheme of the interaction between Ca^2+^ ions and GO flakes; (**d**) stability improvement of GO membranes when intercalated with Ca^2+^ ions. Adapted with permission from [113]. Copyright 2014, the Royal Society of Chemistry.

**Figure 5 membranes-13-00429-f005:**
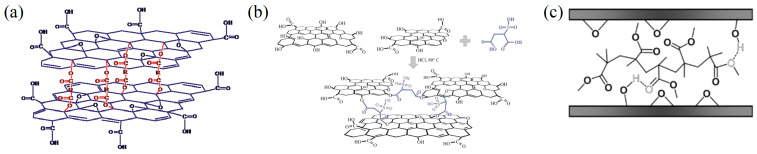
(**a**,**b**) Covalent crosslinking (chemical intercalation). Adapted with permission from [139,140]. Copyright 2013, the Royal Society of Chemistry; (**c**) non-covalent cross-linking (physical intercalation). Adapted with permission from [144], Copyright 2010, Wiley.

**Figure 6 membranes-13-00429-f006:**
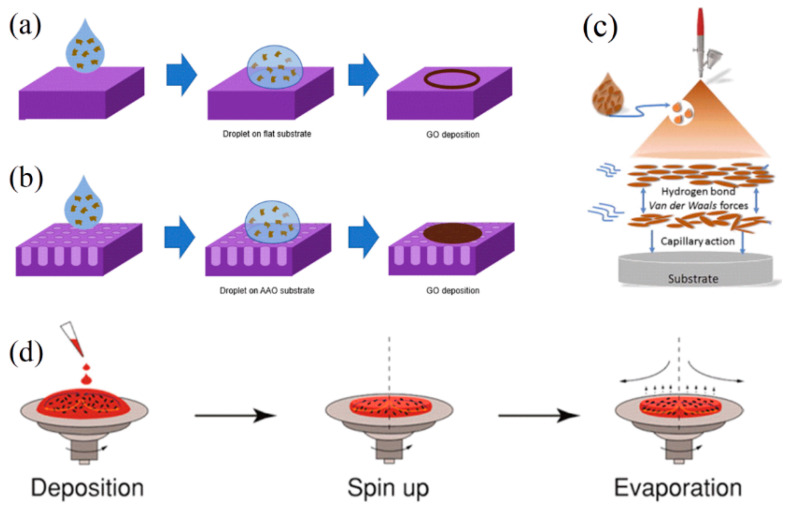
Graphical representation of the preparation of GO membranes by (**a**) drop casting on a flat substrate and (**b**) drop casting on a porous substrate. Adapted with permission from [158]. Copyright 2021, IOP Publishing; (**c**) spray coating. Adapted with permission from [162]. Copyright 2017, Elsevier; (**d**) spin coating. Adapted with permission from [163]. Copyright 2016, American Chemical Society.

**Figure 7 membranes-13-00429-f007:**
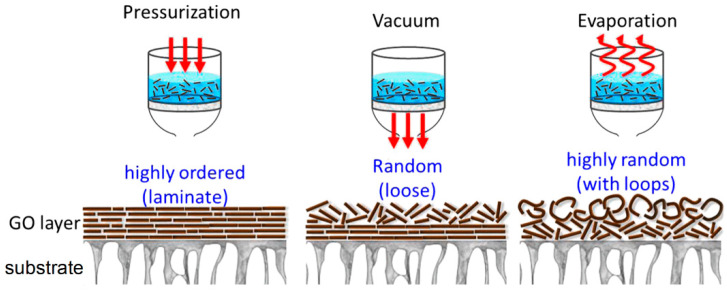
Effect of the microstructure of graphene oxide membranes fabricated through different self-assembly techniques. Adapted with permission from [171]. Copyright 2015, Elsevier.

**Figure 8 membranes-13-00429-f008:**
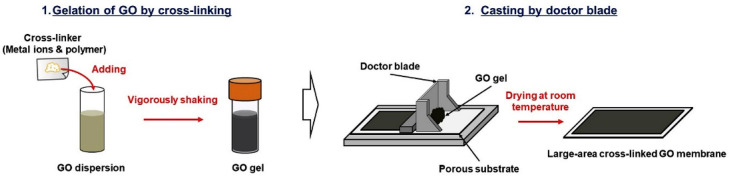
Schematic procedure for the fabrication of large-area crosslinked GO membranes by doctor blade technique. Reproduced with permission from [25]. Copyright 2019, Elsevier.

**Table 1 membranes-13-00429-t001:** Comparison between the main GO membrane fabrication methods according to influential parameters related to the cost and environmental impact of the membranes.

Method	Energy Consumed	Time Consumed	Membrane Order	Available Thickness	Scalability
Drop casting	low	medium	Low	any	yes
Doctor blade	low	medium	Medium	any	yes
Spray coating	low	low	Medium	limited	yes
Spin coating	low	medium	Medium	limited	possibly
Layer-by-layer assembly	high	high	High	limited	no
Vacuum filtration	high	medium	High	limited	no
Pressure assisted filtration	high	medium	Very high	any	no

## Data Availability

No new data were created.

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
