# Peer review of "Green Methods for the Fabrication of Graphene Oxide Membranes: From Graphite to Membranes"

_membranes, 2023, doi:10.3390/membranes13040429_

Round 1

Reviewer 1 Report

A well-written and informative paper

Author Response

We thank the reviewer, we read carefully the manuscript and improved the quality of English in our paper

Reviewer 2 Report

The review written by Alessandro Pedico et al is a well documented paper which covered the production both of graphene oxide and graphene oxide membranes. I think that the paper is very interesting for the readers however i suggest the author to come with more details regarding the functionalization of GO. I would put more accent on covalent and non-covalent functionalization of the GO.

Author Response

We thank the reviewer for this suggestion. We have added more information about GO functionalization and its advantages and applications.

Reviewer 3 Report

The manuscript reviews various methods for fabricating different types of graphene oxide and its membranes, discussing the length of this work is appropriate for its scope in search for greener methods to convert graphite to membranes. Some suggestions are provided below to improve the manuscript.

1) In a short perspective review like this work, it would be useful to provide the authors' view on current techniques to make the nanosheets and the membranes based on existing work and therefore future recommendations to this field. The techniques should also be appropriate for its intended membrane applications (this has not been discussed).

2) The aims and scope are not clear in the Abstract and Introduction. The purpose and the scope should be provided at the end of the introduction.

3) Ball milling has not been discussed- as this is a highly potential green exfoliation approach.

4) Some errors were spotted. For examples, in line 30: "such as distillation and filtration" should be moved to describe the traditional methods; in line 88: "ration"; line 226: "solubility" should read "dispersity"

5) Line 322: "Self-standing membranes.... is needed" It is not necessarily true that free-standing membranes will show an increased flow rate. A supported membrane allows the fabrication of thinner membranes which will give rise to a higher flow rate.

Author Response

  1. In a short perspective review like this work, it would be useful to provide the authors' view on current techniques to make the nanosheets and the membranes based on existing work and therefore future recommendations to this field. The techniques should also be appropriate for its intended membrane applications (this has not been discussed). 

An improved explanation and comparison between graphene oxide fabrication techniques have been added to the manuscript. However, a deep study of the applications of graphene oxide membranes is out of the scope of this review as it has been extensively studied (https://doi.org/10.1016/j.nantod.2018.04.007, https://doi.org/10.1016/j.cclet.2022.01.034, https://doi.org/10.3390/ijms20225609, https://doi.org/10.1016/j.coche.2017.03.002, https://doi.org/10.1016/j.carbon.2018.07.040, https://doi.org/10.1021/acsanm.1c04469, https://doi.org/10.1007/s12274-022-4273-y ) 

  1. The aims and scope are not clear in the Abstract and Introduction. The purpose and the scope should be provided at the end of the introduction. 

We have improved the abstract to better convey the message of the present work. 

  1. Ball milling has not been discussed- as this is a highly potential green exfoliation approach. 

We thank the reviewer for highlighting this missing technique. We have added ball milling as an exfoliation approach in the manuscript. 

  1. Some errors were spotted. For examples, in line 30: "such as distillation and filtration" should be moved to describe the traditional methods; in line 88: "ration"; line 226: "solubility" should read "dispersity" 

We thank the reviewer for highlighting this aspect. We checked the manuscript and have rewritten the sentences and fixed the errors. 

  1. Line 322: "Self-standing membranes.... is needed" It is not necessarily true that free-standing membranes will show an increased flow rate. A supported membrane allows the fabrication of thinner membranes which will give rise to a higher flow rate. 

We completely agree with the reviewer. The message we wanted to convey was that properly designed unsupported membranes can (but not always) increase the flow rate by removing one or more unnecessary layers of materials responsible for pressure drop and hindered flow of permeate. The sentence has been modified accordingly. 

Reviewer 4 Report

1)      Some of the grammatical errors/ sentence formation need to be corrected.

For an example in the conclusion, Low energy processes should be preferred to high temperature or high energy processes to reduce their environmental impact, as the use of green solvents and reagents

2)      The abstract is not clear. It should be clearly stated what is reviewed in this manuscript. It looks like an abstract written to a book chapter.

3)      The following paper should be quoted which deals about some of the aspects of nanofiltration and their application related to this study.

Sundarrajan et al, Desalination 2013, 308, 198-208

Author Response

  1. Some of the grammatical errors/ sentence formation need to be corrected. 

For an example in the conclusion, Low energy processes should be preferred to high temperature or high energy processes to reduce their environmental impact, as the use of green solvents and reagents 

We thank the reviewer for highlighting this aspect. We checked the manuscript and have rewritten the sentences which needed to be modified. 

  1. The abstract is not clear. It should be clearly stated what is reviewed in this manuscript. It looks like an abstract written to a book chapter. 

We have improved the abstract to better convey the message of the present work. 

3)      The following paper should be quoted which deals about some of the aspects of nanofiltration and their application related to this study. 

Sundarrajan et al, Desalination 2013, 308, 198-208 

We thank the reviewer for the suggestion. We added many works to the paper discussing various aspects of GO membranes, their properties and their use.